# Learning continuous-time PDEs from sparse data with graph neural networks

**Valerii Iakovlev, Markus Heinonen & Harri Lähdesmäki**
Department of Computer Science
Aalto University
Helsinki, Finland
`{valerii.iakovlev, markus.o.heinonen, harri.lahdesmaki}@aalto.fi`

## Abstract

The behavior of many dynamical systems follow complex, yet still unknown partial differential equations (PDEs). While several machine learning methods have been proposed to learn PDEs directly from data, previous methods are limited to discrete-time approximations or make the limiting assumption of the observations arriving at regular grids. We propose a general continuous-time differential model for dynamical systems whose governing equations are parameterized by message passing graph neural networks. The model admits arbitrary space and time discretizations, which removes constraints on the locations of observation points and time intervals between the observations. The model is trained with continuous-time adjoint method enabling efficient neural PDE inference. We demonstrate the model's ability to work with unstructured grids, arbitrary time steps, and noisy observations. We compare our method with existing approaches on several well-known physical systems that involve first and higher-order PDEs with state-of-the-art predictive performance.

## 1 Introduction

We consider continuous dynamical systems with a state $u(\mathbf{x}, t) \in \mathbb{R}$ that evolves over time $t \in \mathbb{R}_+$ and spatial locations $\mathbf{x} \in \Omega \subset \mathbb{R}^D$ of a bounded domain $\Omega$. We assume the system is governed by an unknown partial differential equation (PDE)

$$\dot{u}(\mathbf{x}, t) := \frac{du(\mathbf{x}, t)}{dt} = F(\mathbf{x}, u, \nabla_{\mathbf{x}} u, \nabla_{\mathbf{x}}^2 u, \ldots), \tag{1}$$

where the temporal evolution $\dot{u}$ of the system depends on the current state $u$ and its spatial first and higher-order partial derivatives w.r.t. the coordinates $\mathbf{x}$. Such PDE models are the cornerstone of natural sciences, and are widely applicable to modelling of propagative systems, such as behavior of sound waves, fluid dynamics, heat dissipation, weather patterns, disease progression or cellular kinetics (Courant & Hilbert, 2008). Our objective is to learn the differential $F$ from data.

There is a long history of manually deriving mechanistic PDE equations for specific systems (Cajori, 1928), such as the Navier-Stokes fluid dynamics or the Schrödinger's quantum equations, and approximating their solution forward in time numerically (Ames, 2014). These efforts are complemented by data-driven approaches to infer any unknown or latent coefficients in the otherwise known equations (Isakov, 2006; Berg & Nyström, 2017; Santo et al., 2019), or in partially known equations (Freund et al., 2019; Seo & Liu, 2019b; Seo et al., 2020). A series of methods have studied neural proxies of known PDEs for solution acceleration (Lagaris et al., 1998; Raissi et al., 2017; Weinan & Yu, 2018; Sirignano & Spiliopoulos, 2018) or for uncertainty quantification (Khoo et al., 2017).

**Related work.** Recently the pioneering work of Long et al. (2017) proposed a fully non-mechanistic method PDE-Net, where the governing equation $F$ is learned from system snapshot observations as a convolutional neural network (CNN) over the input domain discretised into a spatio-temporal grid. Further works have extended the approach with residual CNNs (Ruthotto & Haber, 2019), symbolic neural networks (Long et al., 2019), high-order autoregressive networks (Geneva & Zabaras, 2020),

and feed-forward networks (Xu et al., 2019). These models are fundamentally limited to discretizing the input domain with a sample-inefficient grid, while they also do not support continuous evolution over time, rendering them unable to handle temporally or spatially sparse or non-uniform observations commonly encountered in realistic applications.

Models such as (Battaglia et al., 2016; Chang et al., 2016; Sanchez-Gonzalez et al., 2018) are related to the interaction networks where object's state evolves as a function of its neighboring objects, which forms dynamic relational graphs instead of grids. In contrast to the dense solution fields of PDEs, these models apply message-passing between small number of moving and interacting objects, which deviates from PDEs that are strictly differential functions.

In Poli et al. (2019) graph neural ordinary differential equations (GNODE) were proposed as a framework for modeling continuous-time signals on graphs. The main limitations of this framework in application to learning PDEs are the lack of spatial information about physical node locations and lack of motivation for why this type of model could be suitable. Our work can be viewed as connecting graph-based continuous-time models with data-driven learning of PDEs in spatial domain through a classical PDE solution technique.

**Contributions.** In this paper we propose to learn free-form, continuous-time, a priori fully unknown PDE model $F$ from sparse data measured on arbitrary timepoints and locations of the coordinate domain $\Omega$ with graph neural networks (GNN). Our contributions are:

- We introduce continuous-time representation and learning of the dynamics of PDE-driven systems
- We propose efficient graph representation of the domain structure using the method of lines with message passing neural networks
- We achieve state-of-the-art learning performance on realistic PDE systems with irregular data, and our model is highly robust to data sparsity

Scripts and data for reproducing the experiments can be found in this github repository.

Table 1: Comparison of machine-learning based PDE learning methods.

| Model | Unknown PDE learning | Continuous time | Free-form spatial domain | Free-form initial/boundary conditions | Reference |
|---|---|---|---|---|---|
| PINN | ✗ | ✓ | ✗ | ✗ | Raissi et al. (2017) |
| AR | ✓ | ✗ | ✗ | ✗ | Geneva & Zabaras (2020) |
| PDE-net | ✓ | ✗ | ✗ | ✓ | Long et al. (2017) |
| DPM | ✗ | ✓ | ✗ | ✓ | Freund et al. (2019) |
| DPGN | ✓ | ✗ | ✓ | ✓ | Seo & Liu (2019b) |
| PA-DGN | ✗ | ✓ | ✓ | ✓ | Seo et al. (2020) |
| Ours | ✓ | ✓ | ✓ | ✓ | |

## 2 METHODS

In this Section we consider the problem of learning the unknown function $F$ from observations $(\mathbf{y}(t_0), \ldots, \mathbf{y}(t_M)) \in \mathbb{R}^{N \times (M+1)}$ of the system's state $\mathbf{u}(t) = (u(\mathbf{x}_1, t), \ldots, u(\mathbf{x}_N, t))^T$ at $N$ arbitrary spatial locations $(\mathbf{x}_1, \ldots, \mathbf{x}_N)$ and at $M+1$ time points $(t_0, \ldots, t_M)$. We introduce efficient graph convolution neural networks surrogates operating over continuous-time to learn PDEs from sparse data. Note that while we consider arbitrarily sampled spatial locations and time points, we do not consider the case of partially observed vectors $\mathbf{y}(t_i)$ i.e. when data at some location is missing at some time point. Partially observed vectors, however, could be accounted by masking the nodes with missing observations when calculating the loss. The function $F$ is assumed to not depend on global values of the spatial coordinates i.e. we assume the system does not contain position-dependent fields (Section 2.1).

We apply the method of lines (MOL) (Schiesser, 2012) to numerically solve Equation 1. The MOL consists of selecting $N$ nodes in $\Omega$ and discretizing spatial derivatives in $F$ at these nodes. We

place the nodes to the observation locations $(\mathbf{x}_1, \ldots, \mathbf{x}_N)$. The discretization leads to $F$ being approximated by $\hat{F}$ and produces the following system of ordinary differential equations (ODEs) whose solution asymptotically approximates the solution of Equation 1

$$\dot{\mathbf{u}}(t) = \begin{pmatrix} \dot{u}_1(t) \\ \vdots \\ \dot{u}_N(t) \end{pmatrix} = \begin{pmatrix} \frac{du(\mathbf{x}_1, t)}{dt} \\ \vdots \\ \frac{du(\mathbf{x}_N, t)}{dt} \end{pmatrix} \approx \begin{pmatrix} \hat{F}(\mathbf{x}_1, \mathbf{x}_{\mathcal{N}(1)}, u_1, u_{\mathcal{N}(1)}) \\ \vdots \\ \hat{F}(\mathbf{x}_N, \mathbf{x}_{\mathcal{N}(N)}, u_N, u_{\mathcal{N}(N)}) \end{pmatrix} \in \mathbb{R}^N. \qquad (2)$$

As the discretized $\hat{F}$ inherits its unknown nature from the true PDE function $F$, we approximate $\hat{F}$ by a learnable neural surrogate function.

The system's state at $\mathbf{x}_i$ is defined as $u_i$, while $\mathcal{N}(i)$ is a set of indices of neighboring nodes other than $i$ that are required to evaluate $\hat{F}$ at $\mathbf{x}_i$, and $\mathbf{x}_{\mathcal{N}(i)}$ with $u_{\mathcal{N}(i)}$ are positions and states of nodes $\mathcal{N}(i)$. This shows that the temporal derivative $\dot{u}_i$ of $u_i$ depends not only on the location and state at the node $i$, but also on locations and states of neighboring nodes, resulting in a locally coupled system of ODEs.

Each ODE in the system follows the solution at a fixed location $\mathbf{x}_i$. Numerous ODE solvers have been proposed (such as Euler and Runge-Kutta solvers) to solve the full system

$$\mathbf{u}(t) = \mathbf{u}(0) + \int_0^t \dot{\mathbf{u}}(\tau)d\tau, \qquad (3)$$

where $0 \leq \tau \leq t$ is a cumulative intermediate time variable. Solving equation 3 forward in time scales linearly both with respect to the number of nodes $N$ and the number of evaluated time points $M$, while saturating the input space $\Omega$ requires a large number of nodes. In practice, PDEs are often applied for two- and three-dimensional spatial systems where the method is efficient.

## 2.1 POSITION-INVARIANT GRAPH NEURAL NETWORK DIFFERENTIAL

After introducing Equation 2, we transition from learning $F$ to learning $\hat{F}$. The value of $\hat{F}$ at a node $i$ must depend only on the nodes $i$ and $\mathcal{N}(i)$. Furthermore, the number of arguments and their order in $\hat{F}$ is not known in advance and might be different for each node. This means that our model $\hat{F}$ must be able to work with an arbitrary number of arguments and must be invariant to permutations of their order. Graph neural networks (GNNs) (Wu et al., 2020) satisfy these requirements. In a more restricted setting, where the number of neighbors and their order is known, (e.g. if the grid is uniform) other types of models such as multilayer perceptrons and convolutional neural networks can be used as well.

We consider a type of GNNs called message passing neural networks (MPNNs) (Gilmer et al., 2017) to represent $\hat{F}$ as

$$\hat{F}_\theta(\mathbf{x}_{\mathcal{N}(i)} - \mathbf{x}_i, u_i, u_{\mathcal{N}(i)}), \qquad (4)$$

where $\mathbf{x}_{\mathcal{N}(i)} - \mathbf{x}_i = \{\mathbf{x}_j - \mathbf{x}_i : j \in \mathcal{N}(i)\}$ and $\theta$ denote parameters of the MPNN.

This formulation assumes the absence of position-dependent quantities in $\hat{F}$, but models based on this formulation are invariant to translations and rotations of $\Omega$, which makes generalization to systems with different node positions feasible, and prevents overfitting by memorizing position-specific dynamics.

We use MPNNs, which is a type of spatial-based GNNs, due to their flexibility and computational efficiency. The main alternative – spectral-based GNNs – have relatively poor scaling with the number of nodes and learn global, or domain-dependent, filters due to the need to perform eigenvalue decomposition of the Laplacian matrix.

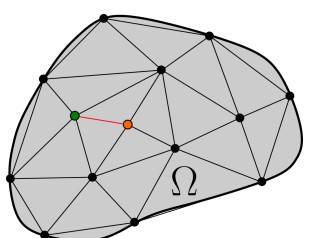

Figure 1: Delaunay triangulation for a set of points. Green and orange points are considered to be neighbors as they share the same edge.

## 2.2 MESSAGE PASSING NEURAL NETWORKS

Let a graph $G = (V, E)$ contain nodes $V = \{\mathbf{x}_i\}_{i=1}^N$, defined by the measurement positions, and undirected edges $E = \{e_{ij}\}$, and

assume each node and edge are associated with a node feature $\mathbf{v}_i$ and an edge feature $\mathbf{e}_{ij}$, respectively. We use the node neighborhood $\mathcal{N}(i)$ to define edges. Neighbors for each node were selected by applying Delaunay triangulation to the measurement positions. Two nodes were considered to be neighbors if they lie on the same edge of at least one triangle (Figure 1). Delaunay triangulation has such useful properties as maximizing the minimum angle within each triangle in the triangulation and containing the nearest neighbor of each node which helps to obtain a good quality discretization of $\Omega$.

In message passing graph neural networks we propagate a latent state for $K \geq 1$ graph layers, where each layer $k$ consists of first aggregating *messages* $\mathbf{m}_i^{(k)}$ for each node $i$, and then updating the corresponding node *states* $\mathbf{h}_i^{(k)}$,

$$\mathbf{m}_i^{(k+1)} = \bigoplus_{j \in \mathcal{N}(i)} \phi^{(k)} \left( \mathbf{h}_i^{(k)}, \mathbf{h}_j^{(k)}, \mathbf{e}_{ij} \right), \tag{5}$$

$$\mathbf{h}_i^{(k+1)} = \gamma^{(k)} \left( \mathbf{h}_i^{(k)}, \mathbf{m}_i^{(k+1)} \right), \tag{6}$$

where $\oplus$ denotes a permutation invariant aggregation function (e.g. sum, mean, max), and $\phi^{(k)}, \gamma^{(k)}$ are differentiable functions parameterized by deep neural networks. At any time $\tau$, we initialise the latent states $\mathbf{h}_i^{(0)} = \mathbf{v}_i = u_i(\tau)$ and node features to the current state $u_i(\tau)$ of the system. We define edge features $\mathbf{e}_{ij} := \mathbf{x}_j - \mathbf{x}_i$ as location differences. Finally, we use the node states at the last graph layer of the MPNN to evaluate the PDE surrogate

$$\frac{d\hat{u}(\mathbf{x}_i, t)}{dt} = \hat{F}_\theta(\mathbf{x}_{\mathcal{N}(i)} - \mathbf{x}_i, u_i, u_{\mathcal{N}(i)}) = \mathbf{h}_i^{(K)}, \tag{7}$$

which is used to solve Equation 3 for the estimated states $\hat{\mathbf{u}}(t) = (\hat{u}(\mathbf{x}_1, t), \dots, \hat{u}(\mathbf{x}_N, t))$.

## 2.3 Adjoint method for learning continuous-time MPNN surrogates

Parameters of $\hat{F}_\theta$ are defined by $\theta$ which is the union of parameters of functions $\phi^{(k)}, \gamma^{(k)}, k = 1, \dots, K$ in the MPNN. We fit $\theta$ by minimizing the mean squared error between the observed states $(\mathbf{y}(t_0), \dots, \mathbf{y}(t_M))$ and the estimated states $(\hat{\mathbf{u}}(t_0), \dots, \hat{\mathbf{u}}(t_M))$,

$$\mathcal{L}(\theta) = \int_{t_0}^{t_M} \ell(t, \hat{\mathbf{u}}) dt = \int_{t_0}^{t_M} \frac{1}{M+1} \sum_{i=0}^{M} ||\hat{\mathbf{u}}(t_i) - \mathbf{y}(t_i)||_2^2 \delta(t - t_i) dt \tag{8}$$

$$= \frac{1}{M+1} \sum_{i=1}^{M} ||\hat{\mathbf{u}}(t_i) - \mathbf{y}(t_i)||_2^2. \tag{9}$$

While discrete-time neural PDE models evaluate the system state only at measurement time points, more accurate continuous-time solution for the estimated state generally requires many more evaluations of the system state. If an adaptive solver is used to obtain the estimated states, the number of time steps performed by the solver might be significantly larger than $M$. The amount of memory required to evaluate the gradient of $\mathcal{L}(\theta)$ by backpropagation scales linearly with the number of solver time steps. This typically makes backpropagation infeasible due to large memory requirements. We use an alternative approach, which allows computing the gradient for memory cost, which is independent from the number of the solver time steps. The approach was presented in Chen et al. (2018) for neural ODEs and is based on the adjoint method (Pontryagin, 2018). The adjoint method consists of a single forward ODE pass 3 until state $\hat{\mathbf{u}}(t_M)$ at the final time $t_M$, and subsequent backward ODE pass solving the gradients. The backward pass is performed by first solving the adjoint equation

$$\dot{\boldsymbol{\lambda}}(t)^T = \frac{\partial \ell}{\partial \hat{\mathbf{u}}(t)} - \boldsymbol{\lambda}(t)^T \frac{\partial \hat{F}}{\partial \hat{\mathbf{u}}(t)}. \tag{10}$$

for the adjoint variables $\boldsymbol{\lambda}$ from $t = t_M$ until $t = 0$ with $\boldsymbol{\lambda}(t_M) = 0$, and then computing

$$\frac{d\mathcal{L}}{d\theta} = -\int_0^T \boldsymbol{\lambda}(t)^T \frac{\partial \hat{F}}{\partial \theta} dt \tag{11}$$

to obtain the final gradient.

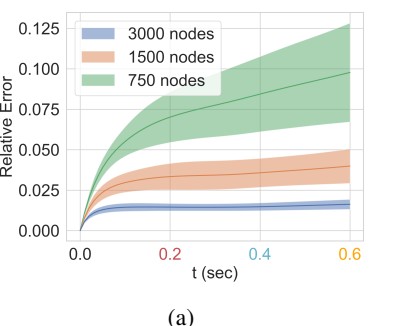 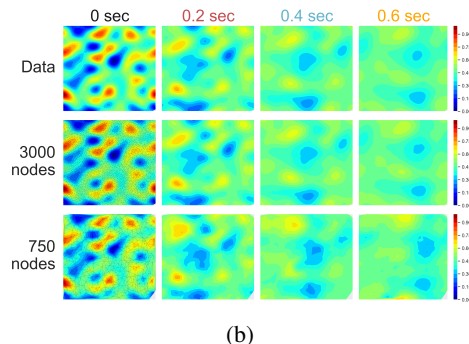

(a)                                                 (b)

Figure 2: a) Relative test errors for different grid sizes. b) Visualization of the true and learned system dynamics (grids are shown in the first column).

## 3 EXPERIMENTS

We evaluate our model's performance in learning the dynamics of known physical systems. We compare to state-of-the-art competing methods, and begin by performing ablation studies to measure how our model's performance depends on measurement grid sizes, interval between observations, irregular sampling, amount of data and amount of noise.

### 3.1 CONVECTION-DIFFUSION ABLATION STUDIES

The convection-diffusion equation is a partial differential equation that can be used to model a variety of physical phenomena related to the transfer of particles, energy, and other physical quantities inside a physical system. The transfer occurs due to two processes: convection and diffusion. The convection-diffusion equation is defined as

$$\frac{\partial u(x,y,t)}{\partial t} = D\nabla^2 u(x,y,t) - \mathbf{v} \cdot \nabla u(x,y,t), \tag{12}$$

where $u$ is the concentration of some quantity of interest (full problem specification and setup are in Appendix A). Quality of the model's predictions was evaluated using the relative error between the observed states $\mathbf{y}(t_i)$ and the estimated states $\hat{\mathbf{u}}(t_i)$:

$$Err = \frac{\|\mathbf{y}(t_i) - \hat{\mathbf{u}}(t_i)\|}{\|\mathbf{y}(t_i)\|}. \tag{13}$$

In all following experiments, unless otherwise stated, the training data contains 24 simulations on the time interval $[0, 0.2]$ sec and the test data contains 50 simulations on the time interval $[0, 0.6]$ sec. The data is randomly downsampled from high fidelity simulations, thus all train and test simulations have different node positions while the number of nodes remains constant. Examples from the train and test sets are shown in Figure 14.

**Different grid sizes.** This experiment tests our model's capability to learn from data with different density of observation points. The time step was set to $0.02$ sec resulting in 11 training time points per simulation. The number of observation points $\mathbf{x}_i$ (and consequently nodes in the GNN) was set to 3000, 1500 and 750. The resulting grids are shown in the first column of Figure 2b. Figure 2 shows relative test errors and models' predictions.

The performance of the model decreases with the number of nodes in the grid. Nonetheless, even with the smallest grid, the model was able to learn a reasonably accurate approximation of the system's dynamics and generalize beyond the training time interval.

**Different measurement time interval.** As will be shown in the following experiments, models with a constant time step are sensitive to the length of the time interval between observations. While showing good performance when the time step is small, such models fail to generalize if the time step is increased. This experiment shows our model's ability to learn from data with relatively large time intervals between observations.

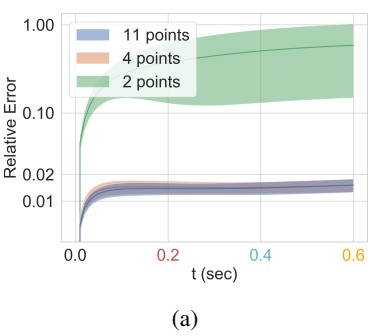

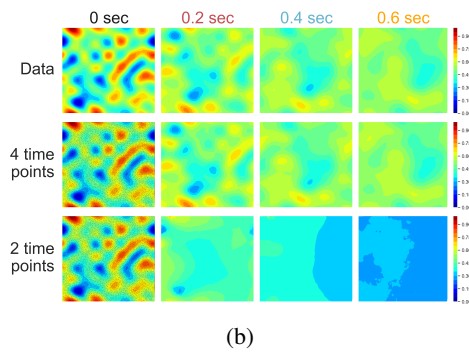

(a)

(b)

Figure 3: a) Relative test errors for different time grids. b) Visualization of the true and learned system dynamics (grids are shown in the first column).

We used 11, 4 and 2 evenly spaced time points for training. The number of nodes was set to 3000. Figure 3 shows relative test errors and models' predictions. The model is able to recover the continuous-time dynamics of the system even when trained with four time point per simulation. Increasing the frequency of observation does not significantly improve the performance. An example of a training simulation with four time points is shown in Figure 11.

**Irregular time step.** Observations used for training might not be recorded with a constant time step. This might cause trouble for models that are built with this assumption. This experiment tests our model's ability to learn from data observed at random points in time.

The model is trained on two time grids. The first time grid has a constant time step 0.02 sec. The second grid is the same as the first one but with each time point perturbed by noise $\epsilon \sim \mathcal{N}(0, (\frac{0.02}{6})^2)$. This gives a time grid with an irregular time step. The time step for test data was set to 0.01 sec. The number of nodes was set to 3000. Relative test errors are shown in Figure 4. In both cases the model achieves similar performance. This demonstrates the continuous-time nature of our model as training and predictions are not restricted to evenly spaced time grids as with most other methods. None of the previous methods that learn free form (i.e., neural network parameterised) PDEs can be trained with data that is sampled irregularly over time.

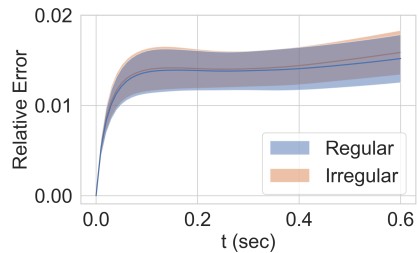

Figure 4: Relative test errors for regular and irregular time grids.

**Different amount of data.** In this experiment, the model is trained on 1, 5, 10 and 24 simulations. The test data contains 50 simulations. The time step was set to 0.01 sec. The number of nodes was set to 3000. Relative test errors are shown in Figure 5. Performance of the model improves as the amount of training data increases. It should be noted that despite using more data, the relative error does not converge to zero.

**Varying amount of additive noise.** We apply additive noise $\epsilon \sim \mathcal{N}(0, \sigma^2)$ to training data with $\sigma$ set to 0.01, 0.02, and 0.04 while the largest magnitude of the observed states is 1. The time step was set to 0.01 sec. The number of nodes was set to 3000. Noise was added only to the training data. The relative test errors are shown in Figure 6. The model's performance decreases as $\sigma$ grows but even at $\sigma = 0.04$ it remains quite high.

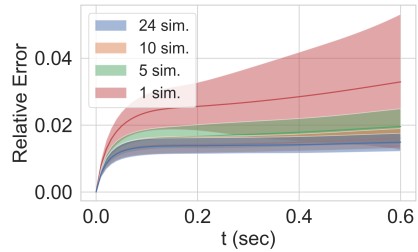

Figure 5: Relative test errors for different amounts of training data.

## 3.2 BENCHMARK METHOD COMPARISON

The proposed model was compared to two models presented in the literature: PDE-Net (Long et al., 2017) and DPGN (Seo & Liu, 2019a). PDE-Net is based on a convolutional neural network and employs a constant time-stepping scheme resembling the Euler method. DPGN is based on a graph neural network and implements time-stepping as an evolution map in the latent space.

We used the PDE-Net implementation provided in Long et al. (2017) except that we pass filter values through an MLP consisting of 2 hidden layers 60 neurons each and `tanh` nonlinearities which helps to improve stability and performance of the model. We use $5 \times 5$ and $3 \times 3$ filters without moment constraints and maximum PDE order set to 4 and 2 respectively. The number of $\delta t$-blocks was set

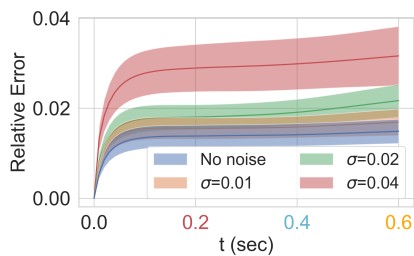

Figure 6: Relative test errors for different amounts of noise in the training data.

to the number of time steps in the training data. Our implementation of DPGN followed that from Seo & Liu (2019a) with latent diffusivity $\alpha = 0.001$. The number of parameters in all models was close to 20k.

The training data contains 24 simulations on the time interval $[0, 0.2]$ sec with the following time steps: 0.01, 0.02 and 0.04 sec. The test data contains 50 simulations on the time interval $[0, 0.6]$ sec with the same time steps. The data was generated on a $50 \times 50$ regular grid as PDE-net cannot be applied to arbitrary spatial grids. Separate models were trained for each time step. The performance of the models was evaluated using the mean of relative test error averaged over time.

Mean relative test errors of the models are shown in Figure 7. The figure shows that performance of the discrete-time models is strongly dependent on the time step while performance of the continuous-time model remains at the same level. At the smallest timestep, PDE-Net with $5 \times 5$ filters outperforms other models due to having access to a

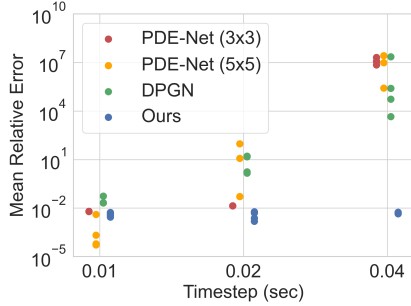

Figure 7: Mean relative errors of models trained with different time steps.

larger neighborhood of nodes which allows the model to make more accurate predictions. However, larger filter size does not improve stability.

We note that some discrete-time models, e.g. DPGN, could be modified to incorporate the time step as their input. Comparison with this type of models would be redundant since Figure 7 already demonstrates the best case performance for such models (when trained and tested with constant time step).

**Importance of relative positional information.** We test our model with and without relative node positions that are encoded as the edge features in our MPNN on grids with a different number of nodes. Smaller number of nodes results in higher distance variability between neighboring nodes (Figure 12) which should increase the dependence of the model accuracy on the relative spatial information. By removing spatial information from our model, we recover GNODE. The models were tested on the heat (Appendix B) and convection-diffusion equations. A full description of the experiment is in Appendix D. The results are shown in Figure 8.

Surprisingly, GNODE shows good results on the purely diffusive heat equation. Nonetheless, the performance of GNODE noticeably differs from that of our model that

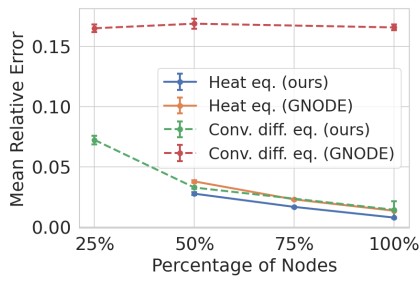

Figure 8: Mean relative test errors of models with and without relative node positions.

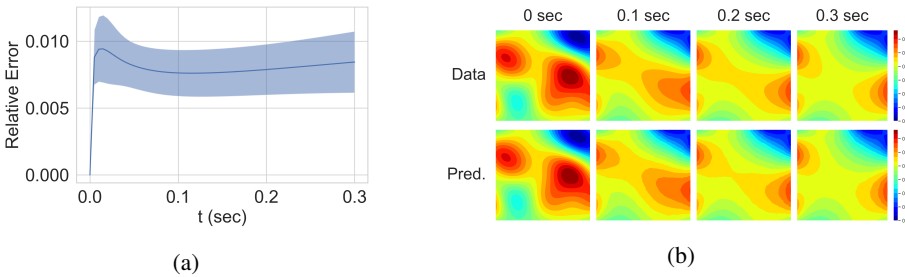

Figure 9: a) Relative test errors for heat equation. b) True and learned system dynamics.

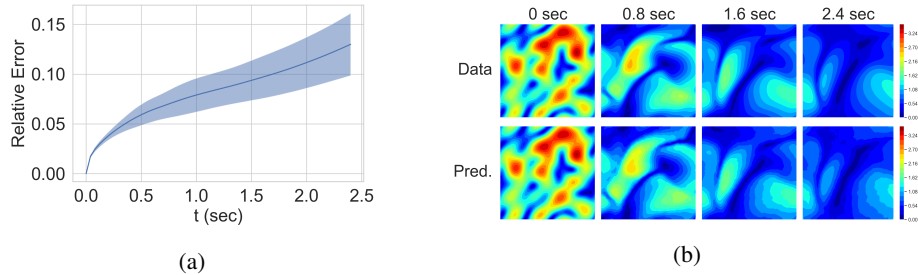

Figure 10: a) Relative test errors for Burgers' equations. b) True and learned system dynamics.

includes the spatial information. Furthermore, the performance difference almost doubles as the number of nodes is decreased from 100% to 50%.

When applied to the convection-diffusion equation, GNODE fail to learn the dynamics irrespective of the number of nodes. This can be explained by the presence of the convective term which transports the field in a specific direction thus making positional information particularly important for accurately predicting changes in the field.

## 3.3 OTHER DYNAMICAL SYSTEMS

The model was tested on two more dynamical systems in order to evaluate its ability to work with a wider range of problems. We selected the heat equation and the Burgers' equations for that purpose. The heat equation is one of the simplest PDEs while the Burgers' equations are more complex than the convection-diffusion equation due to the presence of nonlinear convective terms. The increase in the problems' difficulty allows to trace the change in the model's performance as we move from simpler to more complex dynamics while keeping the number of model parameters fixed.

**Heat equation.**   The heat equation describes the behavior of diffusive systems. The equation is defined as $\frac{\partial u}{\partial t} = D\nabla^2 u$, where $u$ is the temperature field (see Appendix B for details). Figure 9 shows relative errors and model predictions for a random test case. The heat equation describes simpler dynamics than the convection diffusion equation which allowed the model to achieve slightly smaller test errors.

**Burgers' equations.**   The Burgers' equations is a system of two coupled nonlinear PDEs. It describes the behavior of dissipative systems with nonlinear propagation effects. The equations are defined in a vector form as $\frac{\partial \mathbf{u}(x,y,t)}{\partial t} = D\nabla^2 \mathbf{u}(x,y,t) - \mathbf{u}(x,y,t) \cdot \nabla \mathbf{u}(x,y,t)$, where $\mathbf{u}$ is the velocity vector field (see Appendix C for details). For visualization and error measurement purposes, the velocity vector field is converted to a scalar field defined by the velocity magnitude at each node. Figure 10 shows relative errors and model predictions for a random test case.

The Burgers' equations describe more complex dynamics than the previous two cases which is reflected in higher relative test errors. Visual comparison of the true and predicted states shows that the model was able to achieve sufficient accuracy at approximating the unknown dynamics.

## 4 CONCLUSION

We present a continuous-time model of dynamical systems whose behavior is governed by PDEs. The model accurately recovers the system's dynamics even when observation points are sparse and the data is recorded at irregular time intervals. Comparison with discrete-time models reveals the advantage of continuous-time models for datasets with larger time intervals between observations, which is typical for real-world applications where measurements can be either tedious or costly, or both. Discretization of the coordinate domain with the method of lines provides a general modeling framework in which arbitrary surrogate functions can be used for approximating $\hat{F}$. The continuous-time nature of the model enables the use of various time integrators ranging from the Euler method to highly accurate adaptive methods. This allows to optimize the choice of the surrogate function and time integration scheme depending on the structure of the data.

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

## A    Convection-diffusion ablation studies

The convection-diffusion equation is a partial differential equation that can be used to model a variety of physical phenomena related to the transfer of particles, energy, and other physical quantities inside a physical system. The transfer occurs due to two processes: convection and diffusion.

Training and testing data was obtained by solving the following initial-boundary value problem on $\Omega = [0, 2\pi] \times [0, 2\pi]$ with periodic boundary conditions:

$$
\begin{aligned}
\frac{\partial u(x,y,t)}{\partial t} &= D\nabla^2 u(x,y,t) - \mathbf{v} \cdot \nabla u(x,y,t), & (x,y) &\in \Omega, \ t \geq 0, \\
u(x,0,t) &= u(x,2\pi,t), & x &\in [0,2\pi], \ t \geq 0, \\
u(0,y,t) &= u(2\pi,y,t), & y &\in [0,2\pi], \ t \geq 0, \\
u(x,y,0) &= u_0(x,y), & (x,y) &\in \Omega,
\end{aligned}
\tag{14}
$$

where the diffusion coefficient $D$ was set to $0.25$ and the velocity field $\mathbf{v}$ was set to $(5.0, 2.0)^T$. The initial conditions $u_0(x, y)$ were generated as follows:

$$\tilde{u}_0(x, y) = \sum_{k, l = -N}^{N} \lambda_{kl} \cos(kx + ly) + \gamma_{kl} \sin(kx + ly) \tag{15}$$

$$u_0(x, y) = \frac{\tilde{u}_0(x, y) - \min \tilde{u}_0(x, y)}{\max \tilde{u}_0(x, y) - \min \tilde{u}_0(x, y)}, \tag{16}$$

where $N = 4$ and $\lambda_{kl}, \gamma_{kl} \sim \mathcal{N}(0, 1)$. The generated data contains $N_s$ simulations. Each simulation contains values of $u(x, y, t)$ at time points $(t_1, \ldots, t_M)$ and locations $(\mathbf{x}_1, \ldots, \mathbf{x}_N)$, where $\mathbf{x}_n = (x_n, y_n)$. Numerical solutions that represent the true dynamics were obtained using the backward Euler solver with the time step of $0.0002$ seconds on a computational grid with 4100 nodes. Training and testing data used in the following experiments is downsampled from these solutions. Quality of the model's predictions was evaluated using the relative error between the observed states $\mathbf{y}(t_i)$ and the estimated states $\hat{\mathbf{u}}(t_i)$:

$$Err = \frac{\|\mathbf{y}(t_i) - \hat{\mathbf{u}}(t_i)\|}{\|\mathbf{y}(t_i)\|}. \tag{17}$$

The model used for all following experiments contains a single graph layer. The mean was selected as the aggregation function. Functions $\phi^{(1)}(u_i, \cdot)$ and $\gamma^{(1)}(u_i, u_j - u_i, \mathbf{x}_j - \mathbf{x}_i)$ were represented by multilayer perceptrons with 3 hidden layers and hyperbolic tangent activation functions. Input/output sizes for $\phi^{(1)}$ and $\gamma^{(1)}$ were set to 4/40 and 41/1 respectively. The number of hidden neurons was set to 60. This gives approximately 20k trainable parameters.

We followed the implementation of the adjoint method and ODE solvers from `torchdiffeq` Python package (Chen et al., 2018). In all following experiments, adaptive-order implicit Adams solver was used with `rtol` and `atol` set to $1.0 \cdot 10^{-7}$. Rprop (Riedmiller & Braun, 1992) optimizer was used with learning rate set to $1.0 \cdot 10^{-6}$ and batch size set to 24.

## B   HEAT EQUATION EXPERIMENT

Training and testing data was obtained by solving the following initial-boundary value problem on $\Omega = (0, 1) \times (0, 1)$ with Dirichlet boundary conditions:

$$\begin{aligned}
\frac{\partial u(x, y, t)}{\partial t} &= D\nabla^2 u(x, y, t), & (x, y) &\in \Omega, \ t \geq 0, \\
u(x, y, t) &= u_0(x, y), & (x, y) &\in \partial\Omega, \ t \geq 0, \\
u(x, y, 0) &= u_0(x, y), & (x, y) &\in \Omega,
\end{aligned} \tag{18}$$

where $\partial\Omega$ denotes the boundaries of $\Omega$ and diffusion coefficient $D$ was set to $0.2$. The initial conditions $u_0(x, y)$ were generated as follows:

$$\tilde{u}_0(x, y) = \sum_{k, l = -N}^{N} \lambda_{kl} \cos(kx + ly) + \gamma_{kl} \sin(kx + ly) \tag{19}$$

$$u_0(x, y) = \frac{\tilde{u}_0(x, y) - \min \tilde{u}_0(x, y)}{\max \tilde{u}_0(x, y) - \min \tilde{u}_0(x, y)}, \tag{20}$$

where $N = 10$ and $\lambda_{kl}, \gamma_{kl} \sim \mathcal{N}(0, 1)$. The generated data contains $N_s$ simulations. Each simulation contains values of $u(x, y, t)$ at time points $(t_1, \ldots, t_M)$ and locations $(\mathbf{x}_1, \ldots, \mathbf{x}_N)$, where $\mathbf{x}_n = (x_n, y_n)$. Numerical solutions that represent the true dynamics were obtained using the backward Euler solver with the time step of $0.0001$ seconds on a computational grid with 4100 nodes. Training and testing data used in the experiments with the heat equation is downsampled from these solutions.

The model used for all experiments with the heat equation contains a single graph layer. The mean was selected as the aggregation function. Functions $\phi^{(1)}(u_i, \cdot)$ and $\gamma^{(1)}(u_i, u_j - u_i, \mathbf{x}_j - \mathbf{x}_i)$ were represented by multilayer perceptrons with 3 hidden layers and hyperbolic tangent activation functions. Input/output sizes for $\phi^{(1)}$ and $\gamma^{(1)}$ were set to 4/40 and 41/1 respectively. The number of hidden neurons was set to 60. This gives approximately 20k trainable parameters.

We followed the implementation of the adjoint method and ODE solvers from `torchdiffeq` Python package (Chen et al., 2018). In all following experiments, adaptive-order implicit Adams solver was used with `rtol` and `atol` set to $1.0 \cdot 10^{-7}$. Rprop (Riedmiller & Braun, 1992) optimizer was used with learning rate set to $1.0 \cdot 10^{-6}$ and batch size set to 24.

In the experiment, the training data contains 24 simulations on the time interval $[0, 0.1]$ sec with time step 0.005 sec resulting in 21 time point. The test data contains 50 simulations on the time interval $[0, 0.3]$ sec with the same time step. The number of observation points $\mathbf{x}_i$ was set to 4100.

## C BURGERS' EQUATIONS EXPERIMENT

Training and testing data was obtained by solving the following initial-boundary value problem on $\Omega = [0, 2\pi] \times [0, 2\pi]$ with periodic boundary conditions:

$$
\begin{aligned}
\frac{\partial \mathbf{u}(x,y,t)}{\partial t} &= D\nabla^2 \mathbf{u}(x,y,t) - \mathbf{u}(x,y,t) \cdot \nabla \mathbf{u}(x,y,t), & (x,y) \in \Omega,\ t \geq 0, \\
\mathbf{u}(x,0,t) &= \mathbf{u}(x,2\pi,t), & t \geq 0, \\
\mathbf{u}(0,y,t) &= \mathbf{u}(2\pi,y,t), & t \geq 0, \\
\mathbf{u}(x,y,0) &= \mathbf{u}_0(x,y), & (x,y) \in \Omega,\ t = 0,
\end{aligned}
\tag{21}
$$

where the diffusion coefficient $D$ was set to 0.15. The unknown function is now vector-valued. Therefore, the initial conditions $\mathbf{u}_0(x,y)$ for each component were generated as follows:

$$
\tilde{u}_0(x,y) = \sum_{k,l=-N}^{N} \lambda_{kl} \cos(kx + ly) + \gamma_{kl} \sin(kx + ly)
\tag{22}
$$

$$
u_0(x,y) = 6 \times \left( \frac{\tilde{u}_0(x,y) - \min \tilde{u}_0(x,y)}{\max \tilde{u}_0(x,y) - \min \tilde{u}_0(x,y)} - 0.5 \right),
\tag{23}
$$

where $N = 2$ and $\lambda_{kl}, \gamma_{kl} \sim \mathcal{N}(0, 1)$. The generated data contains $N_s$ simulations. Each simulation contains values of $u(x,y,t)$ at time points $(t_1, \ldots, t_M)$ and locations $(\mathbf{x}_1, \ldots, \mathbf{x}_N)$, where $\mathbf{x}_n = (x_n, y_n)$. Numerical solutions that represent the true dynamics were obtained using the backward Euler solver with the time step of 0.0016 seconds on a computational grid with 5446 nodes. Training and testing data used in the experiments with the heat equation is downsampled from these solutions.

The model used for all experiments with the Burgers' equations contains a single graph layer. The mean was selected as the aggregation function. Functions $\phi^{(1)}(u_i, \cdot)$ and $\gamma^{(1)}(u_i, u_j - u_i, \mathbf{x}_j - \mathbf{x}_i)$ were represented by multilayer perceptrons with 3 hidden layers and hyperbolic tangent activation functions. Input/output sizes for $\phi^{(1)}$ and $\gamma^{(1)}$ were set to 6/40 and 41/2 respectively. The number of hidden neurons was set to 60. This gives approximately 20k trainable parameters.

We followed the implementation of the adjoint method and ODE solvers from `torchdiffeq` Python package (Chen et al., 2018). In all following experiments, adaptive-order implicit Adams solver was used with `rtol` and `atol` set to $1.0 \cdot 10^{-7}$. Rprop (Riedmiller & Braun, 1992) optimizer was used with learning rate set to $1.0 \cdot 10^{-6}$ and batch size set to 24.

In the experiment, the training data contains 24 simulations on the time interval $[0, 0.8]$ sec with time step 0.04 sec resulting in 21 time point. The test data contains 50 simulations on the time interval $[0, 2.4]$ sec with the same time step. The number of observation points $\mathbf{x}_i$ was set to 5000.

## D RELATIVE POSITIONAL INFORMATION EXPERIMENT

Data generation, time intervals, models and hyper parameters for this experiment are described in Appendix B for the heat equation, and Appendix A and Section 3.1 for the convection diffusion equation.

For the heat equation, 100% of nodes corresponds to 1000 nodes while for the convection-diffusion equation it corresponds to 3000 nodes. The number of training time points was set to 21 in both cases.

# E EXTRA FIGURES

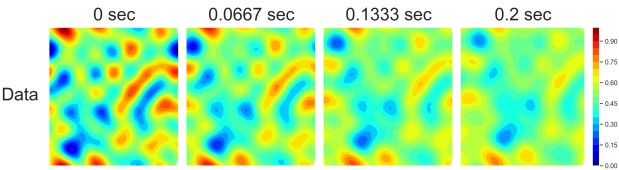

Figure 11: Differences between observations in a train case with 4 time points.

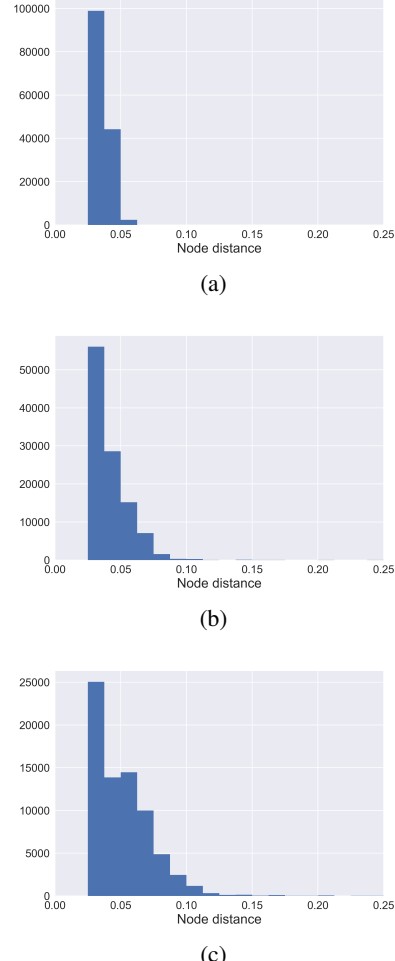

Figure 12: Relative node distances for graphs with different number of nodes. a) 1000 nodes, b) 750 nodes, c) 500 nodes.

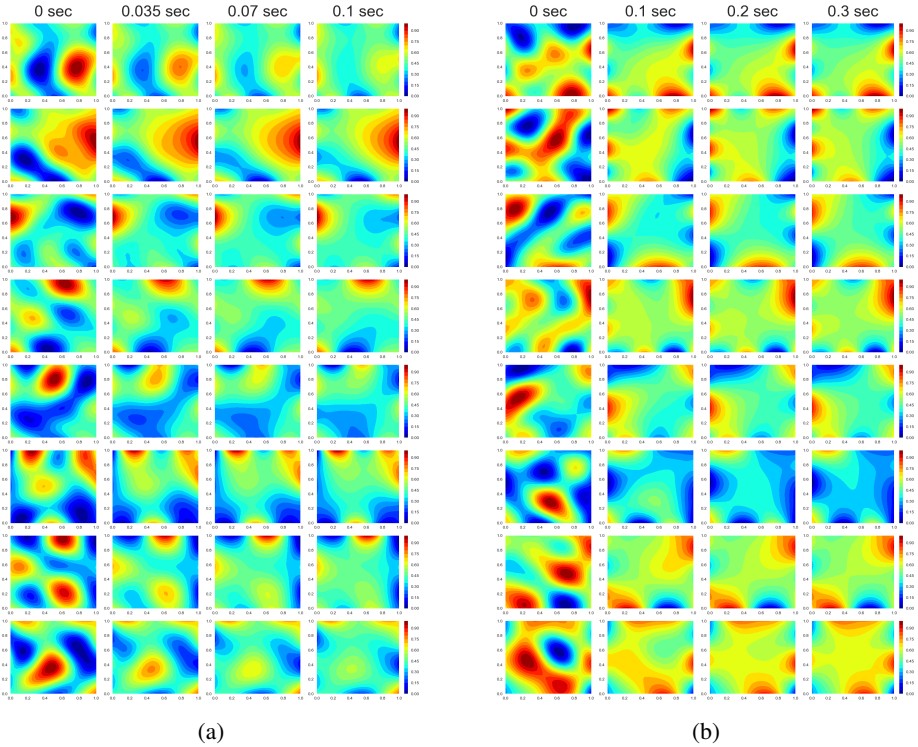

Figure 13: Snapshots of train (a) and test (b) simulations for the heat equation.

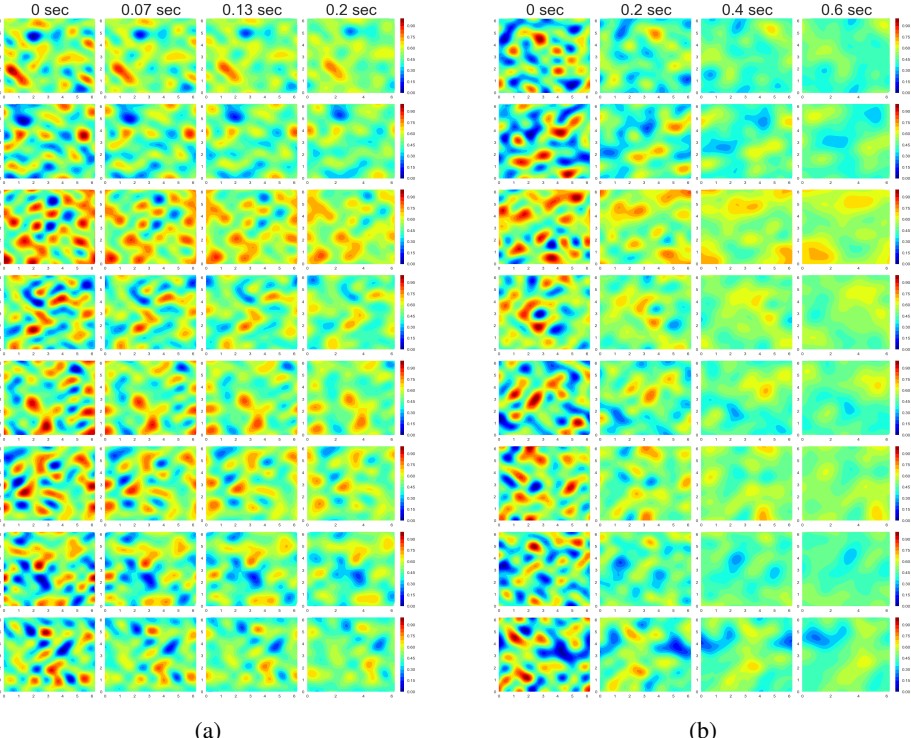

Figure 14: Snapshots of train (a) and test (b) simulations for the convection-diffusion equation.

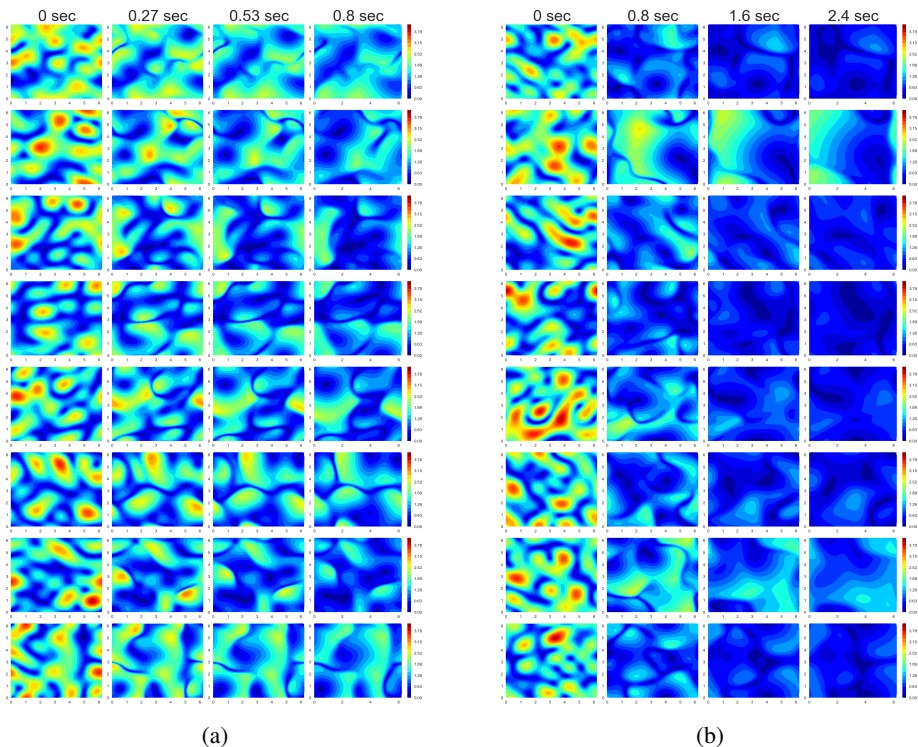

$$(a) \hspace{9cm} (b)$$

Figure 15: Snapshots of train (a) and test (b) simulations for the Burgers' equations.

## F  APPLYING TRAINED MODELS TO GRIDS OF DIFFERENT SIZES

Figure 2b shows grids with different numbers of nodes. The grid with 3000 nodes nodes contains neighborhoods of similar shapes and sizes while neighborhoods in the grid with 750 nodes differ in shapes and sizes over a much larger range. This suggests that models trained on the grid with 750 nodes would work reasonably well on grids with 1500 and 3000 nodes, but not vice versa. We demonstrate this in the table below. The data and models used for this experiments are the same as in Section 3.1.

Table 2: Mean relative errors of models trained on some grid and applied to other grids.

| Model / Grid size | 3000 | 1500 | 750 |
|---|---|---|---|
| **3000** | $0.013 \pm 0.001$ | $0.017 \pm 0.001$ | $0.043 \pm 0.004$ |
| **1500** | $0.050 \pm 0.005$ | $0.032 \pm 0.001$ | $0.036 \pm 0.001$ |
| **750** | $0.142 \pm 0.034$ | $0.086 \pm 0.004$ | $0.073 \pm 0.004$ |

The model trained on 3000 nodes generalizes poorly to coarser grids while the model trained on 750 grids performs fairly well on all grids. The model trained on 750 nodes performs better on test data with 3000 and 1500 nodes than with 750 nodes. This is because the finer grid allows to make more accurate predictions, therefore the error does not grow as large as for the coarse grid with 750 nodes.

