# OpenReview forum: "Learning continuous-time PDEs from sparse data with graph neural networks"
_ICLR.cc/2021/Conference — ICLR 2021 Poster_

### Official Review · AnonReviewer3 · 2020-10-27
**Nice evaluation of graph-based networks for PDEs with open questions on the side of continuous time**

**Rating:** 7
**Confidence:** 4

**Review:**

The paper proposes to use graph-based networks for evaluations of PDEs with continuous time formulations. In contrast to existing works on continuous time ODE formulations with graph structures, the proposed networks incorporate relative spatial information in order for the network to evaluate spatial derivatives in addition to the temporal dynamics. A key argument for this setup is the flexibility in spcae (via the graph nets) in addition to a variable time step.

The proposed setup is evaluated for the relatively simple PDE cases, a convection diffusion case, a pure diffusion case, and another advection diffusion case for transport. Despite the simplicity, the authors make an effort to illustrate the behavior of their method with an ablation study, and to compare to previous work. Here, they compare to PDE-net, which was originally proposed for system identification, and is used for predictions over time instead here. In addition, they compare to the GNODE approach by Poli et al., which omits the spatial information, but already incorporates ODEs into graph nets. For the latter, the authors demonstrate that for simple cases (pure diffusion) the GNODE approach does a good job to identify dynamics purely over time, while including advection terms significantly increases the error without spatial information. This is good to see and makes sense.

For the PDE-net comparison, I was wondering how the 3 time step sizes were incorporated into the PDE-net. Isn't this "by-default" a fixed time step architecture? Was it changed to receive the time step as an input, or does Figure 7 show 3 networks, i.e. one per timestep? The appendix unfortunately does not provide any additional details on how the comparison was executed.

I also have to admit I couldnt follow the argumentation why graph-nets, e.g., from Sanchez-Gonalez et al. '18 aren't suitable in the context of the paper. Sure, they are evaluated on moving positions, but isn't that even more difficult compared to the static locations used here? So when keeping these positions fixed, wouldn't the networks potentially do an even better job than for the moving locations? As a plus, they wouldn't require a Delaunay triangulation or similar meshing step.

While the ablation study is generally nice, I was missing one central component here: as the paper targets continuous time (it's highlighted in the title), I expected an evaluation regarding how much is gained from the introduction of continuous time. The larger time steps of Fig. 3 seem quite trivial, but what I instead hoped to see here was a comparison to a model that simply receives the chosen tilmestep dt as an additional input, and is trained in a supervised fashion with data from a few different time step sizes (i.e. non-continuous, but varying dt). This is maybe what was done for the PDE-net, but the paper is not clear here. I think it would be important to demonstrate the advantages of a continuous formulation, which introduces a significant amount for complexity, over a much simpler training with discrete but varying time steps.

I hope the authors can shed light on this aspect during the rebuttal, as apart from this relatively central open question I like the paper. Thus, assuming that the authors can clarify this part and show that the proposed method yields benefits, I think this could be an interesting paper for ICLR. It presents an interesting evaluation of PDE-learning with graph-nets, which I would consider to be interesting for many ICLR attendees.

---

> ### Author Response · Authors · 2020-11-20
> **Answer to AnonReviewer3**
>
> **Q1**: For the PDE-net comparison, I was wondering how the 3 time step sizes were incorporated into the PDE-net. Isn't this "by-default" a fixed time step architecture?
> **A1**: Indeed, this information should have been included in the experiment description. We used separate models for each time step.
>
> **Q2**: I also have to admit I couldnt follow the argumentation why graph-nets, e.g., from Sanchez-Gonalez et al. '18 aren't suitable in the context of the paper. As a plus, they wouldn't require a Delaunay triangulation or similar meshing step.
> **A2**: The model from Sanchez-Gonzalez et al. perhaps could be adapted to learning PDEs. However, it was designed for physical systems of discrete agents (e.g. cartpole, pendulum, stick figures, etc.), and its main contributions were in modelling the internal structure of the agent (i.e. how the joints of the legs are linked to the joints in torso, etc.). As such, it would require considerable amount of work to adapt it to learning PDEs. We tried to select most suitable models for our comparisons and found DPGN to be a more suitable and readily available graph-based discrete-time model to compare against.
>
> Please note that meshing is a way of finding neighbors for each node and thus is a necessary step for any graph-based model, not only for ours.
>
> The concept of PDEs with moving positions corresponds to the Lagrangian reference frame and is very interesting, but adapting the model to such moving PDE systems is not immediately obvious, and would require more research.
>
> **Q3**: While the ablation study is generally nice, I was missing one central component here: as the paper targets continuous time (it's highlighted in the title), I expected an evaluation regarding how much is gained from the introduction of continuous time. The larger time steps of Fig. 3 seem quite trivial, but what I instead hoped to see here was a comparison to a model that simply receives the chosen tilmestep dt as an additional input, and is trained in a supervised fashion with data from a few different time step sizes (i.e. non-continuous, but varying dt). I think it would be important to demonstrate the advantages of a continuous formulation.
> **A3**: That's a great point. To the best of our knowledge, there are no previous neural network parameterized PDE models that are designed to be trained with a varying time step. This implies that currently available models would need to be augmented to have such capability. This could be done by explicitly adding dt as the input, but the only models that allow to do that are the ones which learn an evolution map (e.g. DPGN). Models that learn dynamics (ours) or mimic integration schemes (PDE-Net) cannot use dt for the model's predictions without changing the model structure.
>
> As for a comparison of our model with a discrete-time model trained on different dts, it was to some extent done in Section 3.2 Figure 7 where DPGN was used with a fixed time step (which is the best case scenario for a discrete model with varying dts). If trained on a varying dt, DPGN would achieve at best similar performance as when trained with a fixed dt but with multiple restrictions. The model would be restricted to work on dts similar to dts from the train set and the maximum dt would have to be below some "critical value" so that the model does not become unstable (e.g. as for dt=0.02 and 0.04 in Figure 7). Continuous-time models, on the other hand, can be trained with arbitrary time steps, as long as they allow to capture the system's dynamics, and can be tested on arbitrary time steps without deterioration of the performance. Furthermore, the "critical value" of dt for such models is much larger than for discrete-time models (Figure 7).
>
> Considering the above, we believe that all information that such an experiment would provide is already contained in the paper. However, we definitely see the need for clarifying this point and adding a similar discussing to the paper.

---

### Official Review · AnonReviewer1 · 2020-10-29

**Rating:** 6
**Confidence:** 3

**Review:**

- Summary

This paper presents a graph neural network model for learning to model PDE dynamical systems. Given its graph structure and its continuous time formulation (employing the adjoint method for differentiation/backpropagation), this method allows the usage of samples placed arbitrarily in space and time.


- Pros

The continuous-time nature of the model allows for the usage of irregular time sample points.

Previously proposed methods either would not work on continuous time, or unstructured grids, or would not be applicable to settings with unknown governing PDEs. This work combines all these features.

The graph-based representation used makes the proposed method invariant to translations and rotations in the spatial domain.



- Cons

Similar graph-based methods that used continuous-time to model differential equation dynamics had been previously presented, e.g. GNODE. The novelty of the proposed method might be limited.

The test cases are simple and the experimental details are somewhat lacking for a full evaluation of the results (more details below in the additional comments).



- Reasons for score

[Edit: Score updated, see discussion below]

Overall, given the "cons" described above, notably the potential lack of strong novelty in the proposed method, and the lacking experimental description and results, I am for now classifying this paper as marginally below the acceptance threshold.
On the positive side, the method seems to perform favorably when compared to other baselines, in comparisons that are actually favorable to the other methods (e.g., using regular grids). Moreover, the method performs well on the tasks it is tested on.
However, I'm concerned with some uncertainties I have regarding the experimental section and the presented results. These are discussed below in the comments.
Moreover, the proposed method centers around using message-passing neural networks to model the differential equation dynamisc. As mentioned above, previous methods had already proposed the usage of graph neural networks with continuous time for the learning of differential equations, and I am not sure that the addition of spatial mesh information to such a graph neural network constitutes a significant enough modification at this point.
Despite the concerns above, I am open to reading the authors' responses and the reviews/comments and changing my opinion depending on how those affect my current uncertainty.



- Additional comments

I believe an important element that is missing from the description of the experiments is a clearer of how much do train and test set actually differ? This would be important to understand how hard the tasks being performed are. Clearly, if training and test set are too similar, the results lose a lot of power.
Morever, since we also dont see any traning vs test plots, it is also hard to see how much performance is different between these two are. (I am not claiming such a plot would be necessary, but merely that given the otherwise lack of information in this direction, it would be helpful information.) I am aware that the appendix includes a description of how the initial conditions for the data are generated, but lacking more information these are hard to grasp intuitively to be able to judge the tasks.

Moreover, the error in model rollouts over time seems to spike at the beginning and then quickly flatten out. It seems strange that errors would spike up initially and then not compound significantly over time. Do the authors have any intuition as to why this is the case? Is it maybe a consequence of the data samples reaching a sort of steady state after some time? If so, wouldn't this weaken the case being made for a continuous-time model?


How would the Delauney triangulation being employed deal with possible obstacles present in the spatial domain? For example, an airfoil might have its opposing boundaries connected by edges (since they are close in space), even though that would supposedly be a solid. Would these types of solids have to be manually specified when extending this method to such scenarios? (This is not a "drawback", of course, it would be expected of most methods that such object boundaries would have to be defined.)


What integrator is used for the experiments?

---

> ### Author Response · Authors · 2020-11-20
> **Answer to AnonReviewer1**
>
> **Q1**: Similar graph-based methods that used continuous-time to model differential equation dynamics had been previously presented, e.g. GNODE. The novelty of the proposed method might be limited.
> **A1**: Indeed, GNODE follows a similar idea of using graph networks to represent interactions between objects. The difference is that the interactions are represented only through the adjacency matrix rather than through relative positions which are crucial for learning PDEs. Furthermore, our method is well motivated as a natural data-driven extension of the method of lines which is a general approach to solving PDEs. We show in "Importance of relative positional information" that our method can significantly improve GNODE in modeling PDEs.
>
> **Q2**: Difference between train and test sets.
> **A2**: Thanks for pointing that out. We agree that it is difficult to grasp the diversity and differences between the train and test sets from the initial conditions alone so we included examples of train and test data in the appendices. The new figures 13-15 show that the dynamics regimes in the train set significantly differ from that in the test set but the model is still able to generalize beyond the training time horizon.
>
> **Q3**: The error in model rollouts over time seems to spike at the beginning and then quickly flatten out. It seems strange that errors would spike up initially and then not compound significantly over time. Do the authors have any intuition as to why this is the case? Is it maybe a consequence of the data samples reaching a sort of steady state after some time?
> **A3**: This is an interesting point. First, the dynamics are slowing down towards the end, but they do not reach a steady state. We included extra plots that show this in the previous answer.
>
> It seems that this error curve behavior is explained by the simulations having much faster changes occurring in the very early phases. For instance, in Figs 13-15 it's clear that the systems are changing a lot in the beginning, while becoming smoother towards the end. We quantified this effect by computing the average difference between two consecutive snapshots of the system's state which shows that the differences have larger magnitude in the beginning, which also implies more challenging fitting problem, and higher errors in the early phases.
>
> The initial spike in the error also comes from the fact that at time t=0 the error is also 0, and the first temporal increment then sees the error jumping upwards to some positive value.
>
> Finally, we want to note that this kind of error plots are commonly reported in other works as well (See [1] fig 3+12, [2] fig 11+20), which also signifies that this is a common behavior in learning PDEs.
>
> **Q4**: How would the Delauney triangulation being employed deal with possible obstacles present in the spatial domain?
> **A4**: You are correct, simple Delaunay triangulation that we use does not consider boundaries and could, for example, connect opposite edges of the airfoil. However, any mesh generator could be used to create a mesh from observation points, so there will not be any problems when switching to the one that is capable of including boundaries.
>
> **Q5**: What integrator is used for the experiments?
> **A5**: We used a variable step size Adams method which is described in "Solving Ordinary Differential Equations I - Nonstiff Problems (E. Hairer)" Chapter III Section 5 and implemented here https://github.com/rtqichen/torchdiffeq/blob/5dbaf8585ff9e601889811b7a5859ccf87dc576a/torchdiffeq/_impl/adams.py.
>
> **References**:
> [1] Long, Zichao, et al. "Pde-net: Learning pdes from data." International Conference on Machine Learning. 2018.
> [2] Geneva, Nicholas, and Nicholas Zabaras. "Modeling the dynamics of PDE systems with physics-constrained deep auto-regressive networks." Journal of Computational Physics 403 (2020): 109056.

---

> > ### Comment · AnonReviewer1 · 2020-11-24
> > **Response**
> >
> > Thank you for your through response.
> >
> > The answers 2 to 5 have clarified all of the respective questions I had made.
> >
> > For question 1, though your answer was clear, I still have slight reservations with respect to the novelty of the paper with respect to GNODE, even after considering your response here, since the contributions described in the answer amount to a simple (though effective) modification to the previous method and a new motivation. Nevertheless, despite the possibly incremental updates with respect to previous work, your results are still stronger than the ones seen in GNODE. Moreover, the other reviewers seem to be content with the novelty of the paper as this was not an issue brought to attention by the others. Therefore, taking all of these factors into account, I will update my score to a "marginally above acceptance threshold".

---

### Official Review · AnonReviewer2 · 2020-10-29
**PDE learning with few constraints**

**Rating:** 6
**Confidence:** 3

**Review:**

This submission proposes extensions of PDE-net that relax some constraints that could help extend the range of applications of this approach. First, rather than fixing a spatial discretization in the form of a grid, the authors use a Delaunay triangulation to represent the domain. The updates to the nodes of this triangulation are performed using a message-passing GNN framework which couples neighboring nodes. Secondly, the authors use a classical adjoint method to allow for arbitrary time-discretizations (though this may be much more expensive in practice).

I am not an expert on the experimental side of this area, but my sense was that the performance of the approach is relatively good, especially compared to other methods when the time step is large. For small time steps, however, PDE-Net is multiple orders of magnitude better than the present approach, a fact that should have been thoroughly discussed in the text rather than just mentioned, because the discrepancy is so large.

All the experimental results are presented as, essentially, measures of the training error. My sense is that the point of this methodology would be to meaningfully extend the time-horizon over which a given PDE could be intergrated, so I would have liked to see how the integrator performed outside the test set.

Sparsity is mentioned in the title and the "contributions" but appears basically nowhere in the main text. I did not actually see in what regard the data was sparse or see a clear theoretical justification as to why the MPNN approach would be superior on sparse data. I suppose the argument is that with arbitrary spatial and time discretization, the method still able to be formulated whereas PDE-Net requires dense spatial discretizations and time discretizations to train. However, the fact that the method could in principle be used on sparse data is not a demonstration that it works on sparse data. How does PDE-Net perform in the 2 and 4 time point cases?

---

> ### Author Response · Authors · 2020-11-20
> **Answer to AnonReviewer2**
>
> **Q1**: For small time steps, however, PDE-Net is multiple orders of magnitude better than the present approach, a fact that should have been thoroughly discussed in the text rather than just mentioned, because the discrepancy is so large.
> **A1**: This is a good point and we agree that a more thorough discussion of the reasons for the observed difference is required. The main reason for this difference is that we used 1-hop neighborhood for GNN-based models while 2-hop neighborhood (5x5 filters) was used for PDE-Net. When trained with 3x3 filters (1-hop neighborhood), performance of PDE-Net drops significantly as shown below.
>
> Model, Average mean relative error +/-std
> PDE-Net (5x5), 0.00111 +/-0.00199
> PDE-Net (3x3), 0.00643 +/-0.00025
> DPGN,          0.02990 +/-0.01708
> Ours,          0.00407 +/-0.00114
>
> We did not use 3x3 filters for PDE-Net in our experiments due to significantly longer training times and absence of experiments with such filters in the original paper [1].
>
> **Q2**: All the experimental results are presented as, essentially, measures of the training error.
> **A2**: No. Please note that all reported errors are test errors, which also includes all compared methods.
>
> **Q3**: My sense is that the point of this methodology would be to meaningfully extend the time-horizon over which a given PDE could be intergrated, so I would have liked to see how the integrator performed outside the test set.
> **A3**: Correct, and that is essentially what we do. The time-horizon could be extended as much as possible as long as the model is sufficiently accurate for the new dynamics regimes and the errors do not compound too much. For testing, we selected the time-horizon to be 3 times larger than for training as a reasonable extension.
>
> For instance, in the convection-diffusion system we train with observations until 0.2 seconds, while we forecast until 0.6 second to the future. The sections 3.1 and 3.2 show that we can accurately predict the 3x horizon.
>
> **Q4**: Sparsity is mentioned in the title and the "contributions" but appears basically nowhere in the main text. How does PDE-Net perform in the 2 and 4 time point cases?
> **A4**: By sparse data we mean that information from some node or time point is missing completely, so we can exclude it which gives arbitrary spatial and temporal grids. That is, our model does not assume observations at regular spatial grids or at regular timepoints, but accepts both arbitrary spatial observation points and arbitrarily spaced (over time) measurements. Our experiments demonstrate applicability of the model in such cases. Yet, sparsity could also arise from missing information at a particular node at a particular time, which would correspond to partially observed state graphs. Extending the model to handle such cases is an interesting direction but showing that the model can work in this setting was not the goal of our work.
>
> Performance of PDE-Net on 2 and 4 time points could be judged from Figure 7 where it was trained on 21, 11, and 5 time points that correspond to sampling intervals of 0.01, 0.02 and 0.04 sec, respectively. At 11 time points (0.02 sec) PDE-Net already becomes unstable and is considerably worse at 5 time points (0.04 sec). This means that PDE-net requires relatively dense data to work well, while our model is robust to much scarcer measurements.
>
> **References**:
> [1] Long, Zichao, et al. "Pde-net: Learning pdes from data." International Conference on Machine Learning. 2018.

---

### Official Review · AnonReviewer4 · 2020-10-31
**Interesting method, disappointing description and discussion**

**Rating:** 7
**Confidence:** 4

**Review:**

Review:

This papers proposes an algorithm to learn a model for spatio temporal data assumed to be described by a stationary spatio temporal PDE.

Data considered in the paper consists of vectors y(t) of observations at time t and a fixed set of spatial indices x.
A model for discrete vector y(t) is proposed in the form of coupled ODEs (one for each x_i) with a sparse coupling arising from a neighbouring graph on spatial inputs x, and sharing the same transition function.
This transition acts on relative local spatial information and on the absolute function values in a neighbourhood.

The continuous time ODE can be solved using classing ODE solver. The model is fit to data using the adjoint method to backprop through the solution given a squared loss.
The ability of the model to learn is evaluated on data generated from PDEs.

+ves:

+ The proposed model is sensible and the paper motivates and describes the method well.
The obtained sparsity of the ODE coupling makes the method scalable. Preserving a continuous time dimension is useful
if data is irregularly sampled

+ Once learned, the model is quite flexible -> adapts to new grids, adapts to various time intervals

Concerns:

- The paper does not describe well the setting in which it is applicable. Spatio temporal data could be observations at random space and time locations. One has to read all the details to understand this while it should be highlighed.
For example, the data arrives in vectors of observations at input (x, t) for different t.
The method would not be able to deal with missing data (a partial vector y(t)) etc.. , the method assumes stationarity.

- The methods cite other approaches that take a different route to the problem (e.g. for example learning the parameters of PDE directly) such approaches have their scaling issues but inherit from half a century of research on approximate solutions to pdes, and come with guarantees. Does this method come with any guarantees? A discussion on this aspect would be useful.


- For infinitesimal time steps, it makes sense to use only local information to build the differential F.
But for longer time steps (think diffusion) you would need information that span further away to get accurate results.
Along these lines, how would a model trained on fine time grid deteriorate as you test it on data with bigger time steps

- I m surprised of the experiment about removing the positional information. Removing it makes no sense when one has stationary PDEs in mind. Same with the noise: if you add noise, performance decreases.



Overall, I like the method and think it can be very useful to members of the community,  but I find the paper lack a broader perspective when describing it.
The PDE connection could be used to discuss intuition on where it works or fails much more and to guide
the model experiments and validation.
Instead, the current writing makes it sound a black box engineering solution has been proposed and is
 tested with no real guiding principle.

For this reason, I am not a big proponent of the paper but do not oppose acceptance.

---

> ### Author Response · Authors · 2020-11-20
> **Answer to AnonReviewer4**
>
> **Q1**: The paper does not describe well the setting in which it is applicable. Partially observed states.
>
> **A1**: Thank you for pointing that out. While we say that arbitrary spatial and time points can be used in the first paragraph of section 2, we agree that all details about the setting in which our method is applicable should be discussed in the same place to aid clarity. Having locally missing information on a single node is out of scope for our work, but this is an interesting open problem to be studied in future. We note that this kind of missing data potentially could be handled in a straightforward manner by masking the nodes with missing information when calculating the loss.
> We will revise our manuscript accordingly.
>
> **Q2**: Other approaches and guarantees.
>
> **A2**: Indeed, some methods, e.g. [2, 1, 3, 4], come with guarantees, but those are different lines of work concerned with either solving known PDEs or approximating solution maps of unknown PDEs. In contrary, the goal of our work is to learn unknown dynamical systems. Since we assume the system is governed by a PDE, it is reasonable to ask how accurately can we approximate the underlying PDE from observations only. In our setting there are two sources of approximation: discretization of the spatial domain, and approximation of the function $\hat{F}$ that defines the dynamics. The former can be addressed by noting that approximation of partial derivatives present in PDEs improves as the grid becomes finer. The latter can be addressed using model-specific guarantees on what kind of functions the model (e.g. MLP, CNN or GNN) can learn.
>
> **Q3**: Local information and generalization to larger time steps.
>
> **A3**: We agree that using information from a larger neighborhood could be beneficial but please note that the time step is not used as our model's input, and our model does not depend on it. Instead, our model learns the (time-invariant) function $\hat{F}$ and uses it to evolve the system's state forward with numerical solvers (eg. Euler or Runge-Kutta), which approximate the continuous-time dynamics. The time step for fixed solvers can be adjusted manually while time step for adaptive solvers is selected automatically. This ensures that the obtained solutions are accurate and the time steps that the solver makes are not too large.
>
> If the forward solver is using too large step sizes, the system evolution will become unreliable. However, we note that our system is built on the assumption of accurate forward solving. Exploring the resiliency of the method to cruder forward solutions (e.g. to save computing resources, to scale to bigger systems) would be an interesting research topic.
>
> **Q4**: Removing positional information and adding noise.
>
> **A4**: We agree on both points. It was important to test the model on noisy data since the function $\hat{F}$ is local, so noisy observations could significantly affect its output. Our results show that our method is indeed not sensitive to noise and thus demonstrate its applicability to real applications with noisy data.
>
> Removing spatial information when having PDEs in mind is indeed not reasonable at all, but it was done as an ablation study and, as was shown, it might not affect some types of PDEs (diffusion) but can have a huge effect for other PDE types (convection). We note that some earlier approaches (eg. the GNODE) do not consider spatial locations, which motivated our ablation study.
>
> **Q5**: PDE connections and choice of experiments.
>
> **A5**: This is a good point. We want to emphasize that our modelling choices were carefully made to match PDE systems to produce a principled model to learn PDEs. That is, we base our model on a classical PDE solution technique; we utilize spatial information; we rely on accurate numerical solvers; the neural architecture is spatially stationary, but non-linear; among others.
>
> Next, due to the model still being a neural network (which can be finicky or have unexpected behavior), we performed a series of ablation studies to really explore the models behavior in various conditions ranging from different grid sizes, number of data, time step irregularities, amounts of noises, etc. We feel that these experiments are important to carve out the limits where our model can be reliably applied to learn realistic PDE from data, and when it starts to struggle (e.g. with too few data). Our experiments can then be seen as our best attempts to open up the black box.
>
> The principles behind our tests are:
>
> Different grid sizes:
> In classical PDE solution techniques the grid size affects how accurate the numerical solution is. Also, it is known that formulas for approximating spatial differentials (e.g. finite differences) become more accurate as the grid gets finer. These suggest that performance of our model should improve as we refine the grid.

---

> > ### Author Response · Authors · 2020-11-20
> > **continued**
> >
> > Different grid structures: most previous methods have assumed that the data is collected from a regular grid, whereas our method works with arbitrary measurement points/grids.
> >
> > Different measurement time intervals:
> > Our method is not affected by different measurement time intervals because our model is continuous-time and thus the system state can be evaluated at any time point. Moreover, our method is the only method that can infer unknown PDEs from arbitrary measurement points collected at arbitrary time points.
> >
> > Varying amount of additive noise:
> > As was mentioned previously, the function $\hat{F}$ considers only its immediate neighbourhood. Therefore, its predictions could be very sensitive to noise as it does not have access to global information that could help to cancel the noise. We reported prediction accuracy for varying amounts of noise.
> >
> > **References**:
> >
> > [1] Shin et al. "On the convergence and generalization of physics informed neural networks." (2020).
> > [2] Raissi et al. "Physics-informed neural networks: A deep learning framework for solving forward and inverse problems involving nonlinear partial differential equations." (2019)
> > [3] Bhattacharya et al. "Model reduction and neural networks for parametric pdes." (2020).
> > [4] Kutyniok et al. "A theoretical analysis of deep neural networks and parametric PDEs." (2019).

---

> > > ### Comment · AnonReviewer4 · 2020-11-23
> > > **response**
> > >
> > > Thanks for the authors' detailed response.
> > > Q3/A3: my bad, misunderstanding on my side.
> > >
> > > Q1/A1: I would have liked to see the updated description of setting and thought application cases,
> > > with clearer specification where it wouldn't work.
> > > Otherwise, the claims feel too broad, and not supported by the manuscript.
> > >
> > > Q5/A5: I might have not conveyed that in my initial review but I feel the key experiment, would be to test transfer of a learned model on some inputs, to new inputs (different grid position, different number of nodes) and not just always retraining on different grids.
> > > Thanks for the clarification regarding the reason for the additive noise experiment.
> > > Just a comment here, but you could use a more principled approach (generative approach as for example in sec 5, neural ode paper) to deal with noise, rather than hoping the model would still fit in the presence of errors.
> > >
> > >
> > > Overall, I like the method but I maintain some of my initial concerns on description
> > > and evaluation (choice of experiments rather than methodology).

---

> > > > ### Author Response · Authors · 2020-11-23
> > > > **response**
> > > >
> > > > Thank you for the clarifications.
> > > >
> > > > **Q1**: I would have liked to see the updated description of setting and thought application cases, with clearer specification where it wouldn't work. Otherwise, the claims feel too broad, and not supported by the manuscript.
> > > > **A1**: We updated the manuscript and added the required comments in green (Section 2 and 3.1). The update in Section 2 includes all the assumptions we make about the data and the dynamical system that we learn.
> > > >
> > > > **Q2**: Q: I feel the key experiment, would be to test transfer of a learned model on some inputs, to new inputs (different grid position, different number of nodes) and not just always retraining on different grids.
> > > > **A2**: Indeed, this is a very important property of the model and this is exactly what we do in all of our experiments. We randomly downsample train and test data from high fidelity simulations so the grids for train and test simulations are different (different node positions, constant number of nodes). We added this detail to experiment descriptions as well.
> > > >
> > > > Our model is not trained for a specific number of nodes and will work well on any grid with neighborhoods similar to the ones on which the model was trained. For example figure 2b shows a grid with 750 nodes. As can be seen, there are neighborhoods of various sizes ranging from large to small ones, so the model trained on grids with 750 nodes should generalize fairly well to grids with 1500 and 3000 nodes (but not vice versa). We demonstrate this in the table below.
> > > >
> > > > | grid\model |  3000  |  1500  |   750  |
> > > > |:----------:|:------:|:------:|:------:|
> > > > |    3000    | 0.0136 | 0.0286| 0.0321 |
> > > > |    1500    | 0.0468 | 0.0322 | 0.0345 |
> > > > |     750    | 0.1201 | 0.0954 | 0.0717 |
> > > >
> > > > Here we take models trained on 3000, 1500 and 750 nodes and evaluate their mean relative errors on test sets with 3000, 1500 and 750 nodes. As expected, the model trained on 3000 nodes generalizes poorly to coarser grids while the model trained on 750 grids performs fairly well on all grids. The reason for that is that grids with 750 nodes contain neighborhoods of various sizes. We believe that this is an important experiment which is missing from our manuscript, so we will add it to the revised version.
> > > >
> > > > The model trained on 750 nodes performs significantly better on test data with 3000 nodes than with 750 nodes. This is because the finer grid allows to make more accurate predictions, therefore the error does not grow as large as for the coarse grid with 750 nodes.

---

> > > > > ### Comment · AnonReviewer4 · 2020-11-23
> > > > > **re-response**
> > > > >
> > > > > Q1/A1, thanks for the highlights
> > > > >
> > > > > Q2/A2, thanks for the prompt response and extra table.
> > > > >
> > > > > I am pleased with the edits and additional results and will update my score.

---

### Decision · Program_Chairs · 2021-01-07
**Final Decision**

**Decision:**

Accept (Poster)

**Comment:**

This paper proposes a new method for learning a model for spatio-temporal data described by an (unknown) spatio-temporal PDE. The model learns a continuous time PDE using the adjunct method and uses graph networks to perform message passing between different discrete time steps on a grid obtained with Delaunay triangulation.

The method initially 3 favorable and 1 unfavorable ratings, but convincing responses to some of the raised issues led to unanimous recommendations for acceptance (not all reviewer feedback after the rebuttal has been made public, but feedback has been made to the privately AC on these issues by different reviewers).

The reviewers appreciated novelty of the method and numerous ablations.

Initially perceived weaknesses were some key experiments on generalization over different grid discretizations; the simplicity of some experiments, and links to different prior art - many of these points have been dealt with by authors in their response.

The AC concurs and proposes acceptance.